# Transmembrane protease serine 2 (*TMPRSS2*) rs75603675, comorbidity, and sex are the primary predictors of COVID-19 severity

Gonzalo Villapalos-García[1],* , Pablo Zubiaur[1,2],* , Rebeca Rivas-Durán[3], Pilar Campos-Norte[3], Cristina Arévalo-Román[3], Marta Fernández-Rico[3], Lucio García-Fraile Fraile[3], Paula Fernández-Campos[1], Paula Soria-Chacartegui[1] , Sara Fernández de Córdoba-Oñate[4], Pablo Delgado-Wicke[4], Elena Fernández-Ruiz[4] , Isidoro González-Álvaro[5] , Jesús Sanz[3], Francisco Abad-Santos[1,2] , Ignacio de los Santos[3]

**By the end of December 2021, coronavirus disease 2019 (COVID-19) produced more than 271 million cases and 5.3 million deaths. Although vaccination is an effective strategy for pandemic control, it is not yet equally available in all countries. Therefore, identification of prognostic biomarkers remains crucial to manage COVID-19 patients. The aim of this study was to evaluate predictors of COVID-19 severity previously proposed. Clinical and demographic characteristics and 120 single-nucleotide polymorphisms were analyzed from 817 patients with COVID-19, who attended the emergency department of the Hospital Universitario de La Princesa during March and April 2020. The main outcome was a modified version of the 7-point World Health Organization (WHO) COVID-19 severity scale (WHOCS); both in the moment of the first hospital examination (WHOCS-1) and of the severest WHOCS score (WHOCS-2). The *TMPRSS2* rs75603675 genotype (OR = 0.586), dyslipidemia (OR = 2.289), sex (OR = 0.586), and the Charlson Comorbidity Index (OR = 1.126) were identified as the main predictors of disease severity. Consequently, these variables might influence COVID-19 severity and could be used as predictors of disease development.**

## Introduction

The severe acute respiratory syndrome coronavirus 2 (SARS-CoV-2) pandemic started in Wuhan, China, in December 2019. This virus causes the new coronavirus disease 2019 (COVID-19). By the end of December 2021, more than 271 million cases and 5.3 million deaths had been reported (1). Although vaccination is a proved effective strategy for pandemic control, it is not yet equally available in all countries of the world (2). Even in some developed countries, the slowdown in vaccination hinders the achievement of herd immunity (3, 4). Hence, reaching worldwide herd immunity seems highly unlikely in the medium to long term, and it is expected that the virus will remain a health problem in the following months and years. Therefore, the authorization of effective therapies and identification of prognostic biomarkers remains crucial to manage COVID-19 patients more rationally. This is of particular importance because strains emerge that may be potentially more infectious, could cause more severe disease and, above all, could escape the protection of vaccines. This could be the case for the emerging strain omicron (B.1.1.529), a novel variant of concern (5).

Although several studies were published to date evaluating genetic biomarkers associated with COVID-19 severity, most were exploratory, showing heterogenic results, and still nowadays, a clinically relevant genetic biomarker was not described. The first were genes involved in virus entrance to the host, such as angiotensin converting enzyme 2 gene (*ACE2*) and transmembrane serine protease 2 gene (*TMPRSS2*). Different research groups have suggested several candidate variants of *ACE2*, namely rs2285666 or rs4646116 (6, 7). On the other hand, for the *TMPRSS2*, variants such as rs2298659, rs17854725, rs12329760, and rs75603675 were found to be different in the frequency in populations more affected by the disease (8, 9). However, a later genome-wide association study (GWAS) of severe COVID-19 with respiratory failure reported two clusters of genes associated with two different polymorphisms: rs11385942 in leucine zipper transcription factor like 1 gene (*LZTFL1*)

[1]Clinical Pharmacology Department, Hospital Universitario La Princesa, Instituto Teófilo Hernando, Universidad Autónoma de Madrid (UAM), Instituto de Investigación Sanitaria La Princesa (IIS-IP), Madrid, Spain [2]Centro de Investigación Biomédica en Red de Enfermedades Hepáticas y Digestivas (CIBERehd), Instituto de Salud Carlos III, Madrid, Spain [3]Infectious Diseases Unit, Hospital Universitario La Princesa, Instituto de Investigación Sanitaria La Princesa (IIS-IP), Madrid, Spain [4]Molecular Biology Unit, Hospital Universitario La Princesa, Instituto de Investigación Sanitaria La Princesa (IIS-IP), Madrid, Spain [5]Rheumatology Service, Hospital Universitario La Princesa, Instituto de Investigación Sanitaria La Princesa (IIS-IP), Madrid, Spain

Correspondence: pablo.zubiaur@salud.madrid.org; francisco.abad@salud.madrid.org; isantosg@salud.madrid.org
*Gonzalo Villapalos-García and Pablo Zubiaur contributed equally to this work.

 

**Table 1.**  World Health Organization COVID-19 score at admission (WHOCS-1), maximum World Health Organization COVID-19 score (WHOCS-2), and Charlson Comorbidity Index (CCI).

| | Male | Female | N | | Male | Female | N |
|---|---|---|---|---|---|---|---|
| WHOCS-1 | | | | WHOCS-2 | | | |
| 1,2 | 96 (45.28%) | 116 (54.72%) | 212 (25.59%) | 1,2 | 96 (44.5%) | 116 (55.5%) | 212 (25.59%) |
| 3 | 88 (53.33%) | 77 (46.67%) | 165 (20.2%) | 3 | 65 (52%) | 60 (48%) | 125 (15.3%) |
| 4 | 263 (61.16%) | 167 (38.84%) | 430 (52.63%) | 4 | 211 (56.42%) | 163 (43.58%) | 374 (45.78%) |
| 5 | 3 (75%) | 1 (25%) | 4 (0.49%) | 5 | 23 (79.31%) | 6 (20.69%) | 29 (3.55%) |
| 6 | 3 (50%) | 3 (50%) | 6 (0.73%) | 6 | 45 (78.95%) | 12 (21.05%) | 57 (6.98%) |
| | | | | 7 | 13 (65%) | 7 (35%) | 20 (2.45%) |
| CCI | | | | | | | |
| | | | | 7 | 15 (62.5%) | 9 (37.5%) | 24 (2.94%) |
| 0 | 72 (49.32%) | 74 (50.68%) | 146 (17.87%) | 8 | 4 (57.14%) | 3 (42.86%) | 7 (0.86%) |
| 1 | 75 (49.67%) | 76 (50.33%) | 151 (18.48%) | 9 | 7 (100%) | 0 (0%) | 7 (0.86%) |
| 2 | 106 (12.97%) | 80 (9.79%) | 186 (22.77%) | 10 | 2 (66.67%) | 1 (33.33%) | 3 (0.37%) |
| 3 | 66 (55.93%) | 52 (44.07%) | 118 (14.44%) | 11 | 1 (100%) | 0 (0%) | 1 (0.12%) |
| 4 | 44 (55.7%) | 35 (44.3%) | 79 (9.67%) | 12 | 0 (0%) | 1 (100%) | 1 (0.12%) |
| 5 | 36 (61.02%) | 23 (38.98%) | 59 (7.22%) | 13 | 2 (100%) | 0 (0%) | 2 (0.24%) |
| 6 | 23 (69.7%) | 10 (30.3%) | 33 (4.04%) | Total | 453 (55.45%) | 364 (44.55%) | 817 (100%) |

and rs657152 in the *ABO* gene (10). Because of the disparity of the observed findings, additional confirmatory and exploratory studies are warranted. The aim of this work was to perform a review of the published single-nucleotide polymorphisms (SNPs) related to COVID-19 prognosis or severity by the end of 2020 and to evaluate them in an independent validation cohort. For this purpose, we genotyped 817 patients managed at Hospital Universitario de La Princesa, for a panel of 120 SNPs selected based on an extensive literature search.

# Results

The population consisted on 817 patients, 453 (55.45%) males and 364 (44.55%) females. The range of age was 19 to 97 yr, where the mean age was 60 yr. The baseline characteristics of the study population are shown in Table 1. Biogeographical origin of patients was inferred by their country of birth: 636 were European, 161 were American, 7 were East Asian, 6 were Near Eastern, and 1 was Central/ South Asian. Most patients were symptomatic and required hospitalization with oxygen supplementation at the moment of the first hospital visit (WHOCS-1 = 4, 51.38%), followed by asymptomatic or mild patients (WHOCS-1 = 1 and 2, 27.92%) and by symptomatic without need for oxygen supplementation (WHOCS-1 = 3, 19.74%). As for the severest clinical situation, 77 died or required ICU admission (WHOCS-2 = 6–7, 9.27%), and the remaining severity groups were distributed similarly like in WHOCS-1. Most patients presented a CCI of 2–8 (93.50%), patients with CCI = 1 accounted for 3.97%, and patients with CCI between 9 and 13 accounted for 2.53% of the population.

Treatments received before the first emergency room visit and during the admission in the hospital are described in Table 2. The

**Table 2.**  Treatments used (a) before first emergency room visit and (b) for the treatment of COVID-19 during admission.

| Before first emergency room visit (n = 190, 23.25%) | |
|---|---|
| ACE inhibitors | 108 (13.22%) |
| Angiotensin II receptor blocker | 123 (15.06%) |
| Aldosterone antagonist | 8 (0.98%) |
| Anticoagulants | 32 (3.92%) |
| Corticoids | 22 (2.69%) |
| Immunosuppressants | 21 (2.57%) |
| For the treatment of COVID-19 (n = 810, 99.14%) | |
| Hydroxychloroquine/chloroquine | 607 (74.30%) |
| Corticoids | 354 (43.33%) |
| Tocilizumab | 107 (13.10%) |
| Heparin | 459 (56.18%) |
| Remdesivir | 8 (0.98%) |
| Lopinavir/ritonavir | 406 (49.69%) |
| Transfusion of hyperimmune plasma | 9 (1.10%) |

most frequently prescribed treatments prior first emergency room were ACE inhibitors (ACEIs) and angiotensin II receptor blockers (ARA-II), received by 13.22% and 15.06% of patients, respectively. The most frequently used treatments for the management of the disease were hydroxychloroquine or chloroquine (74.30%), heparin (56.18%), lopinavir/ritonavir combination (49.69%), and corticoids (43.33%).

The univariate analysis of severity 1 and 2 variables is shown in Table S1, including a summary of nominally significant variables

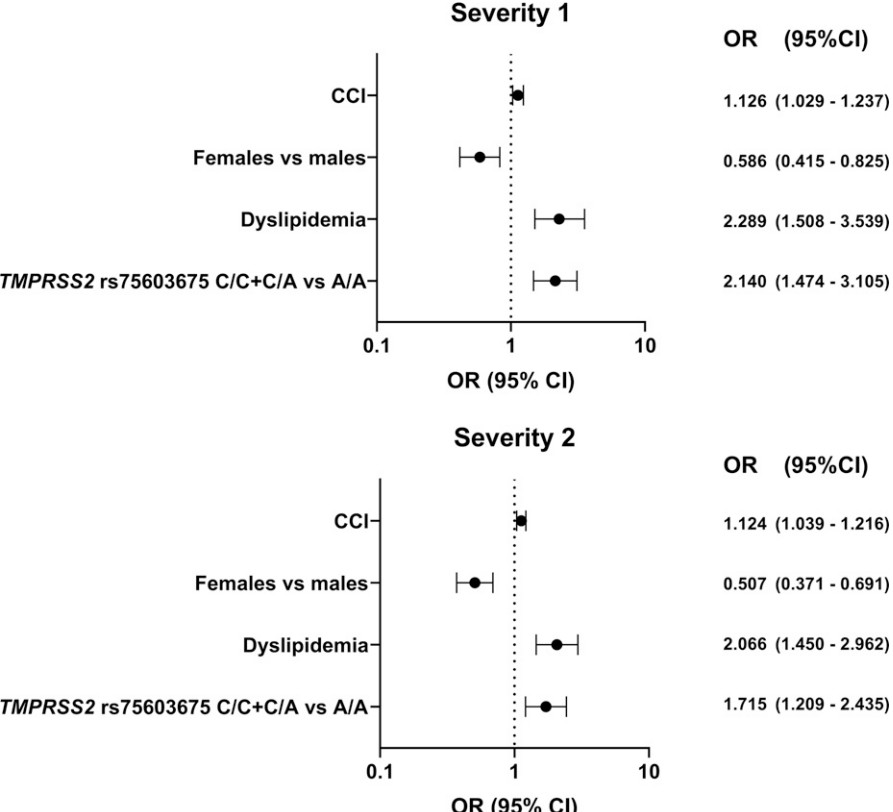

Figure 1. Forest-plot showing statistically significant associations and their odds ratio (OR) between the Charlson Comorbidity Index (CCI), sex, dyslipidemia, and the *TMPRSS2* rs75603675 genotype and COVID-19 severity at the moment of the first hospital emergency room visit (severity 1) and the severest COVID-19 status (severity 2).

and those who reached a corrected *P'* < 0.05, which were included in the multivariate analysis. Biogeographical group resulted non-significant. Males compared with females (OR = 0.586), a higher CCI (OR = 1.126) (covariates), dyslipidemia (OR = 2.289), and *TMPRSS2* rs75603675 (C/C or C/A diplotypes, compared with the A/A diplotype) (OR =2.140) were significantly related to a higher severity 1 and 2 status, after multivariate analysis and Bonferroni correction for multiple comparisons (Fig 1).

The univariate analysis of WHOCS-1 and WHOCS-2 is shown in Table S1, including a summary of nominally significant variables and those who reached a corrected *P'* < 0.05, which were included in the multivariate analysis. The same variables identified in the multivariate analysis of severities 1 and 2 were now observed in the multivariate analysis of WHOCS-1 and WHOCS-2 (Table 3).

Based on the estimates obtained from the multivariate analysis, the following equations were proposed to calculate WHOCS-1 and WHOCS-2 scales in infected in Table 4.

The remaining variables, that is, drugs used before COVID-19 infection, drugs used for the treatment of the disease, the remaining polymorphisms, etc., were unrelated to disease severity in both analyses.

## Discussion

The scientific community's effort to investigate biomarkers for predicting the risk and severity of infection was strenuous since December 2019 to date. A huge amount of works were published in this regard, including reviews and systematic reviews (11). Although there is some consensus on which biomarkers can track disease progression (mainly pro-inflammatory cytokines, ferritin, etc.), there is no biomarker that can predict disease progression from baseline. Baseline health status and demographic characteristics, including sex, age, and comorbidities have been described as the main predisposing factors (12, 13, 14, 15, 16). However, a percentage of severity is not explained by the latter factors (12, 13). Genetic polymorphism may explain part of this susceptibility, which resulted in dozens of publications proposing several SNPs and other genetic alterations associated with susceptibility to COVID-19 infection and severity (Table 5). However, to our knowledge, very few studies validated these associations and their potential clinical relevance. Our intention was, therefore, to design a panel of polymorphisms to validate their usefulness in an independent set of patients.

To prevent bias, first, we proposed a very strict statistical analysis to avoid obtaining spurious results. Second, we proposed to correct for confounding factors in all the statistical tests performed and therefore decided to consider the CCI, which included known COVID-19 severity predictors such as age and obesity, and the sex as covariates. Third, we decided not to analyze some variables dependent on the pandemic situation at the time of recruitment. For example, we did not consider ICU admission as a valid measure of COVID-19 severity as this was restricted because of hospital collapse. In other words, some patients who reached sufficient severity

**Table 3. Multivariate analysis of WHOCS-1 and WHOCS-2 variability.**

| Multivariate | | | | | | | | | |
|---|---|---|---|---|---|---|---|---|---|
| **WHOCS-1** | | | | | **WHOCS-2** | | | | |
| | **Estimate** | **SE** | **P** | **P′** | | **Estimate** | **SE** | **P** | **P′** |
| (Intercept) | 1.962 | 0.156 | <0.001 | <0.001 | (Intercept) | 2.597 | 0.141 | <0.001 | <0.001 |
| Dyslipidemia | 0.579 | 0.137 | <0.001 | <0.001 | Dyslipidemia | 0.551 | 0.124 | <0.001 | <0.001 |
| *TMPRSS2* rs75603675 C/C + C/A versus A/A | 0.591 | 0.138 | <0.001 | <0.001 | *TMPRSS2* rs75603675 C/C + C/A versus A/A | 0.405 | 0.125 | 0.001 | 0.005 |
| CCI | 0.101 | 0.030 | 0.001 | 0.003 | CCI | 0.120 | 0.027 | <0.001 | <0.001 |
| Sex | −0.326 | 0.122 | 0.008 | 0.032 | Sex | -0.415 | 0.111 | <0.001 | 0.001 |

WHOCS-1, modified World Health Organization COVID-19 severity scale at first hospital examination; WHOCS-2, highest score on the modified World Health Organization COVID-19 severity scale; CCI, Charlson Comorbidity Index; SE, standard error; *P*, nominal significance; *P′*, significance after Bonferroni correction for multiple comparisons.

to merit admission to an intensive care unit were not admitted because there was no bed available.

Our findings on the predictors of COVID-19 severity are consistent with previous publications in the literature. The CCI was previously related to COVID-19 prognosis (89, 90), which is consistent with the correlation observed in this work between a higher score and higher WHOCS-1 and -2 scores. Furthermore, males get infected, require ICU admission, mechanical ventilation, and die more frequently than women (91), which is consistent with the protective effect observed here for the female sex regarding WHOCS-1 and more intensely with WHOCS-2. Additional studies are required to determine the underlying differences behind this sexual dimorphism. Moreover, dyslipidemia was previously related to severe COVID-19 prognosis (92), which is consistent with our findings, where the presence of dyslipidemia was related to a 0.579 and 0.551 higher WHOCS-1 or WHOCS-2 scores, respectively. Finally, *TMPRSS2* rs75603675 C/C or C/A diplotypes, compared with the A/A diplotype, were related to 0.591 higher WHOCS-1 and 0.405 higher WHOCS-2 scores, respectively. The contribution of this polymorphism requires further discussion.

The transmembrane serine protease 2, encoded by the *TMPRSS2* gene, participates in several physiological and pathological situations, being up- and down-regulated by several hormonal processes. It is used by several viruses to enter host cells, including the *Influenza* virus and the human coronaviruses HCoV-229E, MERS-CoV, SARS-CoV, and SARS-CoV-2 (93, 94). Genetic polymorphism of this gene was described to affect disease severity. Particularly, the *P*.Val197Met (rs12329760) variant is defined as deleterious and previously reported to have a protective effect on the patients (95). Here, this variant had no effect on WHOCS-1 or -2 variability, whereas the *P*.Gly8Val (rs75603675) missense variant (C > A) was related to a significantly higher WHOCS-1 and WHOCS-2. In contrast, in one study, the prevalence of the *TMPRSS2* rs75603675 A allele was similar between infected patients, compared with uninfected (96). Unfortunately, no information about infection severity is provided in the latter article. Therefore, to the best of our knowledge, this is the first work to propose that this variant has a significant impact on COVID-19 prognosis and severity. Probably, the reason behind this association is explained by the down-regulation of the protein in *TMPRSS2* rs75603675 A allele carriers or the expression of a protein

**Table 4. Equations for the prediction of WHOCS-1 and WHOCS-2 for COVID-19 severity of patients.**

| WHOCS-1 | | | |
|---|---|---|---|
| | | 1.962 | Basal severity |
| + | | 0.579 | If patient has dyslipidemia |
| + | | 0.591 | If *TMPRSS2* rs75603675 C/C or C/A genotypes are present |
| + | | 0.101 | x CCI |
| − | | 0.326 | If patient is female |
| Total: | | | |
| WHOCS-2 | | | |
| | | 2.597 | Basal severity |
| + | | 0.551 | If patient has dyslipidemia |
| + | | 0.405 | If *TMPRSS2* rs75603675 C/C or C/A genotypes are present |
| + | | 0.120 | x CCI |
| − | | 0.415 | If patient is female |
| Total: | | | |

**Table 5.** Genes, single-nucleotide polymorphism (SNPs) identifiers, maximum allele frequencies in Iberians and Americans, and the impact of the reviewed and included variants.

| Gene | SNPs (rs) | MAF IBS | MAF AMR | Variant impact | References |
|---|---|---|---|---|---|
| ABCB1 | rs1045642 | 0.46 (A) | 0.43 (A) | Synonymous variant | 17 |
| | rs1128503 | 0.38 (A) | 0.40 (A) | Synonymous variant | 17 |
| | rs2032582 | 0.41 (A) | 0.37 (A) | Missense variant | 17 |
| | rs2032582 | 0.02 (T) | 0.06 (T) | Missense variant | 17 |
| ABO | rs657152 | 0.37 (A) | 0.30 (A) | Intron variant | 18 |
| ACE | rs4291 | 0.36 (T) | 0.28 (T) | Regulatory region variant | 19 |
| | rs1799752 | Not available | Not available | Intron variant | 20 |
| | rs4343 | 0.44 (A) | 0.39 (G) | Synonymous variant | 7 |
| ACE2 | rs143695310 | 0.04 (A) | 0.03 (A) | Intergenic variant | 21 |
| | rs2106809 | 0.28 (G) | 0.32 (G) | Intron variant | 22 |
| | rs1978124 | 0.43 (T) | 0.29 (T) | Intron variant | 23 |
| | rs5936029 | 0.47 (C) | 0.29 (C) | Intron variant | 21 |
| | rs1996225 | 0.47 (T) | 0.45 (C) | Intron variant | 21 |
| | rs4646156 | 0.34 (A) | 0.25 (A) | Intron variant | 7 |
| | rs2285666 | 0.25 (T) | 0.34 (T) | Splice region variant | 24 |
| | rs2074192 | 0.40 (T) | 0.40 (T) | Intron variant | 7 |
| | rs35803318 | 0.10 (T) | 0.07 (T) | Synonymous variant | 25 |
| | rs4830542 | 0.35 (C) | 0.28 (C) | Intergenic variant | 7 |
| | rs4646116 | 0.01 (C) | <0.01 (C) | Missense variant | 26 |
| | rs4646188 | 0.14 (G) | 0.03 (G) | Intron variant | 27 |
| | rs41303171 | 0.03 (C) | <0.01 (C) | Missense variant | 21 |
| ADAM17 | rs55790676 | 0.24 (T) | 0.13 (T) | 5′ UTR variant | 28 |
| | rs12692386 | 0.30 (G) | 0.48 (G) | 5′ UTR variant | 29 |
| AGT | rs699 | 0.58 (G) | 0.37 (A) | Missense variant | 30 |
| ApoE | rs7412 | 0.06 (T) | 0.05 (T) | Missense variant | 31 |
| | rs429358 | 0.14 (C) | 0.10 (C) | Missense variant | 31 |
| CCL2 | rs1024611 | 0.29 (G) | 0.49 (G) | Regulatory region variant | 32 |
| CCL5 | rs2107538 | 0.12 (T) | 0.23 (T) | 5′ UTR variant | 32 |
| CD14 | rs2569190 | 0.49 (G) | 0.47 (G) | Intron variant | 32 |
| CD147 | rs8259 | 0.33 (A) | 0.38 (A) | 3′ UTR variant | 33 |
| CD69 | rs11052877 | 0.36 (G) | 0.34 (G) | 3′ UTR variant | 34 |
| CLEC2D | rs1560011 | 0.41 (A) | 0.36 (A) | Intron variant | 34 |
| CRP | rs1130864 | 0.32 (A) | 0.33 (A) | 3′ UTR variant | 35 |
| CSF3 | rs2227322 | 0.40 (G) | 0.33 (G) | 5′ UTR variant | 36 |
| CYP2C19 | rs12248560 | 0.21 (T) | 0.12 (T) | Intron variant | 17 |
| | rs4244285 | 0.15 (A) | 0.10 (A) | Synonymous variant | 17 |
| CYP2C9 | rs1799853 | 0.14 (T) | 0.10 (T) | Missense variant | 17 |
| | rs1057910 | 0.08 (C) | 0.04 (C) | Missense variant | 17 |
| CYP3A4 | rs67666821 | <0.01 (T x 6) | <0.01 (T x 6) | Frameshift variant | 17 |
| | rs35599367 | 0.04 (A) | 0.03 (A) | Intron variant | 17 |
| CYP3A5 | rs776746 | 0.08 (T) | 0.20 (T) | Splice acceptor variant | 17 |
| CYP4V2 | rs13146272 | 0.40 (A) | 0.47 (A) | Missense variant | 37 |

| Gene | SNPs (rs) | MAF IBS | MAF AMR | Variant impact | References |
|---|---|---|---|---|---|
| DPP4 | rs56179129 | 0.01 (T) | 0.01 (T) | Missense variant | 9, 38, and 39 |
| | rs116302758 | 0.01 (C) | 0.016 (C) | Splice acceptor variant | 9, 38, and 39 |
| | rs17574 | 0.38(G) | 0.17(G) | Missense variant | 9, 38, and 39 |
| | rs17848916 | 0.01(C) | 0.03 (C) | Intron variant | 9, 38, and 39 |
| ENOX1 | rs9594987 | 0.48 (C) | 0.49 (C) | Intron variant | 34 |
| EPHX1 | rs1051740 | 0.29 (C) | 0.32 (C) | Missense variant | 40 |
| F11 | rs2289252 | 0.41 (T) | 0.34 (T) | Noncoding transcript exon variant | 41 |
| | rs2036914 | 0.46 (T) | 0.46 (T) | Intron variant | 42 |
| FGG | rs2066865 | 0.22 (A) | 0.21 (A) | Intergenic variant | 43 |
| G6PD | rs1050828 | <0.01 (T) | 0.01 (T) | Missense variant | 17 |
| | rs1050829 | 0.01 (C) | 0.03 (C) | Missense variant | 17 |
| | rs5030868 | <0.01 (A) | <0.01 (A) | Missense variant | 17 |
| GC (DBP) | rs7041 | 0.43 (A) | 0.46 (A) | Missense variant | 44 |
| | rs4588 | 0.27 (T) | 0.20 (T) | Missense variant | 44 |
| HCP5 | rs2395029 | 0.02 (G) | 0.02 (G) | Noncoding transcript exon variant | 45 |
| HO-1 | rs2071746 | 0.42 (T) | 0.31 (T) | Intron variant | 46 |
| IFITM3 | rs12252 | 0.03 (G) | 0.18 (G) | Splice region variant | 47, 48, and 49 |
| IFNL3 | rs12979860 | 0.30 (T) | 0.40 (T) | Intron variant | 17 |
| IL10 | rs1800896 | 0.41 (C) | 0.30 (C) | Intron variant | 32 |
| | rs1800871 | 0.26 (A) | 0.33 (A) | 5' UTR variant | 32 |
| IL13 | rs1800925 | 0.18 (T) | 0.23 (T) | Noncoding transcript exon variant | 50 |
| IL17A | rs3819025 | 0.07 (A) | 0.34 (A) | Intron variant | 51 and 52 Preprint |
| | rs2275913 | 0.07 (A) | 0.21 (A) | Intergenic variant | 53 |
| IL1B | rs1143627 | 0.32 (G) | 0.45 (A) | 5' UTR variant | 50 |
| | rs1143634 | 0.20 (A) | 0.13 (A) | Synonymous variant | 50 |
| IL1RN | rs315952 | 0.25 (C) | 0.21 (C) | Missense variant | 9 |
| IL6 | rs1800795 | 0.35 (C) | 0.18 (C) | Intron variant | 50, 54, 55, and 56 |
| | rs1800796 | 0.05 (C) | 0.30 (C) | Noncoding transcript exon variant | 32, 54, and 57 |
| | rs1818879 | 0.33 (A) | 0.48 (A) | Regulatory region variant | 58 |
| IL6R | rs2228145 | 0.40 (C) | 0.46 (A) | Missense variant | 59 |
| | rs4329505 | 0.15 (C) | 0.12 (C) | Intron variant | 17 |
| | rs7529229 | 0.37 (T) | 0.41 (T) | Intron variant | 60 |
| | rs11265618 | 0.16 (T) | 0.13 (T) | Intron variant | 17 |
| | rs12083537 | 0.20 (G) | 0.16 (G) | Intron variant | 17 |
| INFL4 | rs11322783 | 0.30 (T) | 0.40 (T) | Frameshift variant | 61 |
| Intergenic region | rs10108210 | 0.49 (C) | 0.28 (C) | Intron variant | 62 |
| | rs703297 | 0.48 (T) | 0.36 (C) | Regulatory region variant | 62 |
| KCNMB1 | rs703505 | 0.38 (G) | 0.47 (G) | Intron variant | 62 |
| LZTFL1 | rs35044562 | 0.05 (G) | 0.05 (G) | Intron variant | 63 |
| MTHFR | rs1801131 | 0.27 (G) | 0.15 (G) | Missense variant | 64 |
| | rs1801133 | 0.44 (A) | 0.47 (A) | Missense variant | 64 |
| NFKB | rs28362491 | 0.42 (-) | 0.49 (-) | Noncoding transcript exon variant | 65 |
| NLRP3 | rs10754555 | 0.39 (G) | 0.42 (G) | Intron variant | 66 and 67 |

| Gene | SNPs (rs) | MAF IBS | MAF AMR | Variant impact | References |
|---|---|---|---|---|---|
| *PEAR1* | rs12041331 | 0.15 (A) | 0.20 (A) | Intron variant | 68 |
| *PTGS1* | rs10306114 | 0.06 (G) | 0.03 (G) | Regulatory region variant | 69 |
| *SLCO1B1* | rs4149056 | 0.12 (C) | 0.13 (C) | Missense variant | 17 and 70 |
| *TLR1* | rs5743551 | 0.29 (C) | 0.47 (C) | Intron variant | 32 |
| *TLR2* | rs1898830 | 0.29 (G) | 0.50 (G) | Intron variant | 32 |
| | rs7656411 | 0.36 (G) | 0.24 (G) | Regulatory region variant | 32 |
| | rs11938228 | 0.33 (A) | 0.49 (C) | Intron variant | 71 |
| | rs3804099 | 0.43 (C) | 0.33 (C) | Synonymous variant | 72 |
| | rs1816702 | 0.08 (T) | 0.16 (T) | Noncoding transcript exon variant | 73 and 74 |
| *TLR4* | rs1927911 | 0.22 (A) | 0.33 (A) | Intron variant | 32 |
| *TLR4* | rs5030728 | 0.27 (A) | 0.24 (A) | Intron variant | 75 and 76 |
| *TLR9* | rs187084 | 0.43 (G) | 0.43 (G) | Intron variant | 32 |
| | rs352162 | 0.45 (T) | 0.46 (T) | Noncoding transcript exon variant | 32 |
| *TMPRSS2* | rs55964536 | 0.50 (T) | 0.29 (T) | Intron variant | 32 |
| | rs383510 | 0.50 (T) | 0.38 (T) | Intron variant | 32, 77, and 78 |
| | rs464397 | 0.47 (T) | 0.29 (T) | Noncoding transcript exon variant | 78 |
| | rs463727 | 0.49 (A) | 0.27 (A) | Intergenic variant | 32 |
| | rs713400 | 0.12 (T) | 0.11 (T) | 5' UTR variant | 79 |
| | rs8134378 | 0.09 (A) | 0.05 (A) | Intron variant | 80 |
| | rs469390 | 0.36 (G) | 0.45 (A) | Missense variant | 78 |
| | rs734056 | 0.47 (C) | 0.32 (A) | Intron variant | 32 |
| | rs2070788 | 0.50 (G) | 0.49 (G) | Intron variant | 32, 77, and 78 |
| | rs12329760 | 0.18 (T) | 0.15 (T) | Missense variant | 8, 79, and 81 |
| | rs77675406 | 0.08 (A) | 0.09 (A) | 3' UTR variant | 79 |
| | rs75603675 | 0.38 (A) | 0.27 (A) | Missense variant | 9 |
| *TNF* | rs1800610 | 0.13 (A) | 0.18 (A) | Intron variant | 82 |
| | rs1799964 | 0.18 (C) | 0.20 (C) | Regulatory region variant | 83 |
| | rs361525 | 0.06 (A) | 0.08 (A) | Regulatory region variant | 84 |
| *TNF/TNKA* | rs1800629 | 0.15 (A) | 0.07 (A) | Regulatory region variant | 32 and 54 |
| *TNKA* | rs1800630 | 0.13 (A) | 0.13 (A) | Regulatory region variant | 32 |
| *TRAF3IP2* | rs13190932 | 0.07 (G) | 0.06 (G) | Missense variant | 85 |
| | rs33980500 | 0.08 (T) | 0.12 (T) | Missense variant | 62 |
| | rs13196377 | 0.06 (G) | 0.07 (G) | Intron variant | 86 |
| *VDR* | rs2228570 | 0.33 (A) | 0.48 (A) | Start lost | 32 |

MAF, maximum allele frequency; IBS, Iberians; AMR, Americans, UTR, untranslated region. Frequency data were obtained from Ensembl (87) and dbSNP (88).

with a structural change that causes a less specific or impaired interaction with viral proteins, causing a less efficient internalization of viruses inside human cells. This is congruent with Latini et al observations: *TPMRSS2* rs75603675 (C > A) A allele was in significantly lower frequency in populations that suffered more severe cases of COVID-19 (57). This suggested a possible protective effect toward COVID-19 infection.

Other nominally significant associations were established between WHOCS scales and: *IFNL4* rs12979860, ACEIs, obesity, ARA-II, *HCP5* rs2395029, *ACE* rs1799752, *DPP4* rs17574, *HMOX1* rs2071746, *IL10*

rs1800896, *IL6* rs1818879, *NFKB1* rs28362491, and *MX1* rs469390. Although some of these associations were previously observed, others were also persistently rejected. For instance, the use of renin-angiotensin-aldosterone system (RAAS) inhibitors was proposed to cause the up-regulation of the ACE2 receptor, the receptor for SARS-CoV-2 protein S, which enables cell infection; therefore, the use of this drugs was related to a higher risk for COVID-19 infection and worse prognosis (97, 98, 99, 100, 101, 102, 103). However, a bias was identified in this assumption as it were the conditions associated with the use of RAAS inhibitors (e.g., hypertension or

heart disease) which actually led to a higher risk of worse prognosis (104). Consequently, we decided to use a sufficiently strict statistical approximation to control for this bias. Hence, we considered all associations mentioned above as negative, and not worth discussing, because this would contribute to confusion.

One noteworthy disparity with our study is the GWAS study by the Severe COVID-19 GWAS Group (18). They report different genetic markers to ours. The explanation that we can elucidate to address this discrepancy is that our outcome variable is different. Our analysis focus in all COVID-19 severity stages, whereas they focus in the most severe cases. Thus, our findings may explain the susceptibility to infection in the early stages of the disease, whereas other factors such as 3p21.31 gene cluster and the ABO blood-group system may determine terminal status of COVID-19. Definitely, more research must be performed in other to clarify the true effect of the *TMPRSS2* rs75603675 polymorphism in the outcome of this disease.

With this work, we integrate multiple clinical factors and genetic factors to COVID-19 severity prediction. A variable that excellently captures the comorbidity of patients is used, the CCI index. The presence or absence of dyslipidemia complements the CCI index. Only with this information, for instance, a significantly higher WHOCS-1 and -2 scores can be predicted for a patient with dyslipidemia and a CCI of five versus a patient without dyslipidemia and CCI of 1. Although this was previously known, that is, patients with higher comorbidities are related to worse COVID-19 prognosis, our scale signifies a clinically relevant tool to better manage patients because it is clear and numerical. Furthermore, sex and *TMPRSS2* rs75603675 stand as additional clinically relevant predictors of COVID-19 severity, which were included in the scales. Continuing with the previous example, if the first patient carried the C/C diplotype and was male and the second patient was female and carried the A/A diplotype, the predicted WHOCS-1 and WHOCS-2 scales would be: 3.637 and 4.153 for the first patient and 1.737 and 2.302 for the second. These predictions are related to specific clinical requirements (e.g., hospitalization or oxygen supplementation). This quantitative measurement could help in the optimization of clinical resources.

Despite the merits of this work, it presents limitations that should be considered. First, patients were recruited from the first COVID-19 wave, which might be considered more than a limitation; it can be considered a strength. As this work was carried with patients infected during March–April 2020, the circulating strains were different from those circulating now. In this sense, the conclusions regarding *TMPRSS2* rs75603675 should be confirmed in the currently circulating strains. Nonetheless, new strains, such as the omicron variant, could be more infectious or pathogenic, being the underlying mechanism of such pathogenicity an enhanced interaction between viral antigens and TMPRSS2. This interaction could be affected by genetic variants of this gene and cause an even greater difference of severity with new strains. Nevertheless, new studies shall confirm the relevance of *TMPRSS2* rs75603675 in new circulating variants. Another limitation is that hospital protocols were severely affected by the emergency healthcare situation of the first COVID-19 wave, and adequacy of the therapeutic effort was needed. This, in combination with the retrospective nature of the study produces relative scarcity of severe and asymptomatic individuals. Consequently, the severity distribution might be skewed. In addition, the nature of the WHOCS-1 and WHOCS-2 analysis by means of the

generalized linear model assumes a normal distribution of these variables. To avoid bias, we performed a strict statistical analysis, in exchange for assuming greater type II error. Moreover, the fact that the exact same variables related to severity 1 and 2 variables were identified is reassuring, along with the strict control for type-1 error.

## Conclusions

The *TMPRSS2* rs75603675 C/C or C/A genotypes compared with A/A, males compared with females, the presence of dyslipidemia, and a higher CCI score were associated with more severe COVID-19, at the first hospital visit and at the most severe point of disease progression. The integration of all these variables into the proposed equations could be a useful clinical tool for the rational management of patients with COVID-19. To our knowledge, this is the first work to propose a similar tool that integrates genetic data capable of predicting the prognosis of COVID-19.

# Materials and Methods

## Study design and participants

This study was designed with an observational and retrospective approach. A total of 817 patients with COVID-19, who attended the emergency department of the Hospital Universitario de La Princesa between 29 March and 29 April 2020, were recruited. Both inpatients and outpatients were considered. Patients were recruited consecutively according to their first visit to the emergency department. To avoid imbalances in the proportions of the severity groups because of the retrospective and consecutive nature of recruitment, checkpoints were applied to prioritize the selection of under-represented patients, that is, severe and mild patients. The first checkpoint was performed at the beginning of May 2021, with 617 patients recruited: 83 (13.5%) mild, 466 (75.5%) moderate, and 68 (11%) severe; and the second was performed at the end of May 2021, with 743 patients: 159 (21.5%) mild, 502 (67.5%) moderate, and 82 (11%) severe. The Ethics Research Committee of Hospital Universitario de La Princesa approved the study protocol. All subjects provided informed consent, except for the deceased. They were scheduled for sampling at the Department of Internal Medicine of Hospital Universitario de La Princesa; stored samples were retrieved from the deceased patients. Research compiled with Spanish and European legislation on biomedical research and with the revised Declaration of Helsinki.

## Variables

Hospital and primary care medical records were used to retrieve disease and clinical information. The main outcome (dependent variable) was a modified version of the 7-point World Health Organization (WHO) COVID-19 severity scale (WHOCS) (105). Briefly, individuals are classified according to the following COVID-19 severity groups: (1) infected, asymptomatic; (2) symptomatic not requiring hospitalization; (3) COVID-19 requiring hospitalization without oxygen supplementation; (4) oxygen supplementation with mask or nasal prongs; (5) noninvasive ventilation or high flow

oxygen; (6) intubation and mechanical ventilation in an intensive care unit (ICU); and (7) death. This scale was evaluated in the moment of the first hospital examination (WHOCS-1) and of the severest WHOCS score (WHOCS-2). For this work, WHOCS levels were grouped as follows: levels 1–2 were considered mild severity, levels 3–4 were considered moderate severity, and levels 5 or greater were considered severe COVID-19. The resulting variables were named "severity-1" and "severity-2," that is, a transformed variable of the severity at admission and at the worst severity status.

As independent variables or covariates, the following ones were analyzed: demographic characteristics (age and sex), comorbidity (obesity, dyslipidemia, tobacco or alcohol use, and the Charlson Comorbidity Index, CCI), biogeographical group (106) (inferred from the country of birth), relevant drugs used before COVID-19 infection, that is, angiotensin converter enzyme inhibitors (ACEIs) (e.g., enalapril, lisinopril), angiotensin receptor II antagonists (ARA-II) (e.g., losartan, irbesartan), aldosterone antagonists (e.g., eplerenone, spironolactone), oral anticoagulants (e.g., acenocoumarol, dabigatran), systemic corticosteroids (e.g., dexamethasone) or systemic immunosuppressants (e.g., tacrolimus or methotrexate), drugs for COVID-19 treatment (hydroxychloroquine or chloroquine, remdesivir, corticosteroids, tocilizumab, heparin, lopinavir/ritonavir, plasma transfusion), and SNPs.

### Genotyping

Published articles evaluating SNPs and COVID-19 were addressed since January 2020 until December 2020. Moreover, articles evaluating polymorphism in important genes related to COVID-19 (e.g., *ACE2*, *ACE*, *IL-6*, or *IFNs*) or the coagulation cascade (e.g., *F11*, *CRP*) were screened however not necessarily published during the pandemic. Finally, variants related to safety and effectiveness of the drugs used for the treatment of the disease were included, that is, pharmacogenetic variants. Sample collection occurred in a period after discharge of the patients, during the months of January to May 2021. Patients who gave informed consent provided 5 ml of blood collected in an EDTA K2 tube. For deceased patients, samples were retrieved from stored collection. Genomic DNA was extracted from peripheral blood samples with a Maxwell RSC automated DNA extractor (Promega) following the manufacturer's instructions. For genotype analysis, a QuantStudio 12K flex thermal cycler along with an OpenArray thermal block was used (Thermo Fisher Scientific). A customized genotyping array was designed with variants shown in Table 5. The references justifying the inclusion of these variants are included.

### Statistical analysis

An online tool for sample size calculation was used (Sample Size Calculator available at https://riskcalc.org/samplesize/). The study was a retrospective observational cohort study. The $\alpha$ or type-1 error rate was set at 0.05, and the power, or 1-$\beta$, was set at 0.9. Exposure was defined as the presence or absence of one or several SNPs related to severe COVID-19. The event to be analyzed was severe COVID-19, defined as the event, that is, WHO COVID-19 severities 5, 6, and 7. Depending on the SNP prevalence, the k value (unexposed to exposed ratio) was set. For SNPs with low prevalence (e.g., 5%), k = 20. For SNPs with 20% prevalence, $k$ = 4. For SNPs with 40% prevalence, k = 1.5. Assuming a probability of the event in the

unexposed group ($P0$) of 0.4 (40%) and of 0.55 (55%) in the risk group ($P1$), the following sample sizes were required: 2,667 for k = 20 (127 exposed and 2,540 unexposed patients), 760 for k = 4 (127 exposed and 608 unexposed patients), and 510 for k = 1.5 (204 exposed and 306 unexposed). Therefore, a sample size between 510 and 2,667 was considered. Assuming greater differences (e.g., $P0$ = 0.3 and $P$ = 0.7), significantly lower sample sizes were required. Finally, the sample size was determined according to the capacity of the genotyping platform, the available budget, and the latter estimations. A genotyping array containing 120 SNPs with a capacity for 920 samples was designed and purchased. Full coverage of the 920 samples genotyping capacity could not be achieved because of failures or the need for repetitions.

Statistical analysis was conducted in R (107). To analyze severity 1 and 2 variables, a univariate analysis was initially performed by ordinal logistic regression with the MASS package (108). As independent variables, the following ones were explored: genetic variables (all SNPs were transformed into *dummy* variables, that is, grouping heterozygous subjects with the most frequent homozygous diplotype), biogeographic group, previous treatments and other comorbidities (e.g., dyslipidemia or tobacco use); CCI and sex were included as covariates, and the level of significance was adjusted based on the Bonferroni correction for multiple comparisons. Those variables with corrected $P'$ < 0.05 were introduced as independent variables in a multiple logistic regression analysis of severities 1 and 2, that is, the multivariate analysis. In this analysis, significance was similarly adjusted based on the Bonferroni correction for multiple comparisons.

A secondary analysis of WHOCS-1 and WHOCS-2 was performed to provide a predictive equation of disease severity. For the univariate analysis, a generalized lineal model was performed by means of individual linear regression, with the CCI and sex as covariates and applying the Bonferroni correction for multiple comparisons. Those variables with $P'$ < 0.05 were introduced as independent variables in the multivariate analysis, in this case, by means of a generalized lineal model with CCI and sex as covariates again.

## Availability of Data and Materials

The datasets used and/or analyzed during the current study are available from the corresponding author on reasonable request.

## Supplementary Information

## Acknowledgments

We would like to acknowledge Belén Gutiérrez Calvo and Ana Fuentes Valera for their contribution in blood sample extraction for this project; Rosa Carracedo Rodríguez for her contribution in DNA extraction and storage and Emilia Roy-Vallejo for her assistance in DNA sample identification. This research received funding from the Community of Madrid through the COVID-19

grants of the year 2020, Fondo Supera COVID-19 from Banco de Santander and CRUE (grant Predinmun-COVID) to I de los Santos, I González-Álvaro, and E Fernández-Ruiz and Instituto de Salud Carlos III (ISCIII) from the Spanish Ministry of Science Innovation and Universities and the European Regional Development Fund (ISCIII-FEDER) "A way to achieve Europe." G Villapalos-García is supported by a PFIS predoctoral grant (FI20/00090), and P Zubiaur's contract with CIBERehd is financed by the "Infraestructura de Medicina de Precisión asociada a la Ciencia y Tecnología (IMPaCT, IMP/00009)" (ISCIII). S Fernández de Córdoba-Oñate and P Delgado-Wicke were supported by Predinmun-COVID and PI19/00096 grants, respectively.

## Author Contributions

G Villapalos-Garcia: conceptualization, resources, data curation, formal analysis, supervision, investigation, visualization, methodology, and writing—original draft, review, and editing.
P Zubiaur: conceptualization, resources, data curation, formal analysis, supervision, investigation, visualization, methodology, project administration, and writing—original draft, review, and editing.
R Rivas-Duran: data curation, investigation, and writing—review and editing.
P Campos-Norte: data curation, investigation, and writing—review and editing.
C Arevalo-Roman: data curation, investigation, and writing—review and editing.
M Fernandez-Rico: data curation, investigation, and writing—review and editing.
L Garcia-Fraile Fraile: conceptualization, data curation, visualization, methodology, and writing—review and editing.
P Fernadez-Campos: investigation and writing—review and editing.
P Soria-Chacartegui: investigation and writing—review and editing.
S Fernandez de Cordoba-Onate: resources and writing—review and editing.
P Delgado-Wicke: resources and writing—review and editing.
E Fernandez-Ruiz: resources and writing—review and editing.
I Gonzalez-Alvaro: resources, methodology, and writing—review and editing.
J Sanz: resources and writing—review and editing.
F Abad-Santos: conceptualization, resources, supervision, funding acquisition, investigation, visualization, methodology, project administration, and writing—review and editing.
I de los Santos: conceptualization, resources, supervision, funding acquisition, investigation, visualization, methodology, project administration, and writing—review and editing.

## Conflict of Interest Statement

F Abad-Santos has been consultant or investigator in clinical trials sponsored by the following pharmaceutical companies: Abbott, Alter, Chemo, Cinfa, FAES, Farmalíder, Ferrer, GlaxoSmithKline, Galenicum, Gilead, Italfarmaco, Janssen-Cilag, Kern, Normon, Novartis, Servier, Silver Pharma, Teva, and Zambon. I de los Santos has received grants from Gilead, ViiV, and Janssen. The remaining authors declare no conflicts of interest.

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
