## [Reviewer comments · Life Science Alliance]

Life Science Alliance

TMPRSS2 rs75603675, comorbidity and sex are the primary predictors of Covid-19 severity.

Gonzalo Villapalos-García, Pablo Zubiaur, Rebeca Rivas-Durán, Pilar Campos-Norte, Cristina Arévalo-Román, Marta Fernández-Rico, Lucio García-Fraile Fraile, Paula Fernández Campos, Paula Soria-Chacartegui, Sara Fernández de Córdoba-Oñate, Pablo Delgado-Wicke, Elena Fernández-Ruiz, Isidoro González-Álvaro, Jesús Sanz, Francisco Abad-Santos, Ignacio de los Santos

DOI: <https://doi.org/10.26508/lsa.202201396>

Corresponding author(s): Dr. Pablo Zubiaur (Clinical Pharmacology Department, Hospital Universitario La Princesa); Dr. Ignacio de los Santos (Infectious Diseases Unit, Hospital Universitario La Princesa)

Review Timeline:

Submission Date:	2022-01-31
Editorial Decision:	2022-03-03
Revision Received:	2022-04-06
Editorial Decision:	2022-05-05
Revision Received:	2022-05-12
Accepted:	2022-05-13

Scientific Editor: Novella Guidi

Transaction Report:

March 3, 2022

Re: Life Science Alliance manuscript #LSA-2022-01396-T

Pablo Zubiaur

Clinical Pharmacology Department, Hospital Universitario La Princesa, Instituto Teofilo Hernando, Universidad Autonoma de Madrid (UAM), Instituto de Investigacion Sanitaria La Princesa (IIS-IP), CIBERehd, Madrid, Spain

Dear Dr. Zubiaur,

Thank you for submitting your manuscript entitled "Transmembrane protease serine 2 (TMPRSS2) rs75603675, comorbidity and sex are the primary predictors of Covid-19 severity." to Life Science Alliance. The manuscript was assessed by expert reviewers, whose comments are appended to this letter. We, thus, encourage you to submit a revised version of the manuscript back to LSA that responds to all the reviewers' points.

Thank you for this interesting contribution to Life Science Alliance. We are looking forward to receiving your revised manuscript.

Sincerely,

B. MANUSCRIPT ORGANIZATION AND FORMATTING:

Reviewer #1 (Comments to the Authors (Required)):

This paper analyzed genetic variants from 817 patients with COVID-19 with data on the severity of the disease. They found a genetic variant, dyslipidemia, and the Charlson Comorbidity Index as the main predictors of the disease. Several previous studies evaluated genetic markers associated with COVID-19 severity, but this paper aimed to show their clinical relevance.

Abstract

The aim of the study should be stated more clearly in the abstract. Additionally, a conclusion should be formed after listing the most important results.

Introduction

When stating "Although several studies were published to date evaluating genetic biomarkers associated with Covid-19 severity" cite relevant papers and mention some examples, if possible.

Methods

"Checkpoints were applied during recruitment to prioritize the selection of under-represented patients" should be explained in detail. Which patients do you consider underrepresented and why is it relevant to the study? Do you mean race, ethnic background, or socioeconomic status? Did you add this information as covariate to the statistical analysis? What are the checkpoints during recruitment?

Please add more details on sample collection. Did you retrieve samples from the patients at the Intensive Care Unit when they arrived?

Results

Please add more details on the individuals. In the methods section, you mention allele frequencies for the SNPs for Hispanics and Iberians. However, you do not explain the ethnic/racial background of the individuals in the results section. Moreover, based on the different allele frequencies in individuals with different ancestry, even if the genetic ancestry is not used, as it is not measured in this study, the racial background could be included as covariate. The significant results might be due to the differing allele frequencies between the different ancestry groups (Iberians are of European, and Americans are of Hispanic (admixed) ancestry). Please state both as limitations in the limitations section.

Please add odds ratios to the text to see the direction of the effects when explaining the results of Figure 1.

Discussion

Do you think that the different variants (delta, omicron, etc.) since you finished sample collection (April 2020) would have different predictors? Please discuss the possibility after "Firstly, patients were recruited from the first Covid-19 wave".

Please explain what the original finding with TMPRSS2 was, in the manuscript where you collected the information on the gene from.

In general, the manuscript is easy to read, but please try to form more technical sentences on some occasions, e.g., "With this work, we link several pieces of the puzzle in place".

Reviewer #2 (Comments to the Authors (Required)):

The authors provide an interesting analysis. I have only one issue that requires clarification:

Page 10, line 181: "Males compared to females, a higher CCI (covariates), dyslipidemia and TMPRSS2 rs75603675 (C/C or C/A diplotypes, compared to the A/A diplotype) were significantly related to a higher Severity 1 and 2 status". This finding is not reflected in Figure 1, where it appears as if males have a lower severity status.

Minor points:

Table 2 and 4: The abbreviations WHOCS-1 and 2 are used, whereas in Figure 1 Severity 1 and 2 is used. This is confusing.

Table 5: the brackets for (0.101xCCI) can be removed as this is confusing

page 15, line 279: generalized linear model

Reviewer #3 (Comments to the Authors (Required)):

In this work the authors perform a targeted SNP analysis based on their reading of the literature and apply it to 817 patients from a single institution in Spain.

They associate a TMPRSS2 SNP with an excess risk of severe disease.

I have expertise in TMPRSS2 but limited expertise in GWAS studies and the statistical methods needed to interpret these data so my comments may be of limited utility.

The authors are to be commended for their work and efforts at a very difficult time.

I wanted to raise some issues:

- 1) The absence of a control group from the same community - I note that the authors quote Iberian/US frequencies for the SNPs of interest - were these detected using the same platform. Is the lack of direct comparison by this group a concern?
- 2) The relatively small number of individuals with severe disease - 106 if use groups 5-7. Given the large number of experimental treatments used in this cohort there appears to be significant potential to introduce bias. I would suggest comments from someone with real biostatistical expertise but the combination of no experimental controls, targeted SNP analysis and a relatively small 'severe' cohort does raise concerns regarding false positives.
- 3) If I understand correctly age and obesity did not predict outcome but sex did. This is contrary to much larger clinical series e.g. the OpenSAFELY study - Williamson et al 2020. Can the authors explain why? Does this again raise concerns regarding the cohort?
- 4) There have been many many studies of SNPs in COVID including some of severity, including one from Madrid: DOI: 10.1056/NEJMoa2020283 They did not implicate this SNP. Was it tested for. How do the authors explain the prior lack of signal at this locus?
- 5) There are a number of minor issues including the labelling of the charts regarding male-female risk which to my reading suggest an excess risk for female sex as presented whereas the text is clear that the opposite is true.

Reviewer #1

1. Abstract: The aim of the study should be stated more clearly in the abstract. Additionally, a conclusion should be formed after listing the most important results.

Answer: thank you very much for your suggestion. We have expanded the abstract to better reflect the results, and included a sentence that captures the conclusion of the study.

Original version (line 41):

“By the end of December 2021, coronavirus disease 2019 (Covid-19) produced more than 271 million cases and 5.3 million deaths. Although vaccination is proved an effective strategy for pandemic control, it is not yet equally available in all countries over the world. Therefore, the authorization of effective therapies and identification of prognostic biomarkers remains crucial to manage Covid-19 patients more rationally. Clinical and demographic characteristics and 120 single nucleotide polymorphisms were analyzed from 817 patients with Covid-19, who attended the emergency department of the Hospital Universitario de La Princesa (Madrid, Spain) during March and April 2020. The main outcome was a modified version of the 7-point world health organization (WHO) Covid-19 severity scale (WHOCS); both in the moment of the first hospital examination (WHOCS-1) and of the severest WHOCS score (WHOCS-2). *TMPRSS2* rs75603675 genotype, dyslipidemia, sex and the Charlson Comorbidity Index were identified as the main predictors of disease severity. Afterwards, the integration of all these variables into simple equations was performed.”

Modified version (line 41):

“By the end of December 2021, coronavirus disease 2019 (Covid-19) produced more than 271 million cases and 5.3 million deaths. **Although vaccination was an effective strategy for pandemic control, it is not yet equally available in all countries. Therefore, identification of prognostic biomarkers remains crucial to manage Covid-19 patients. The aim of this study was to evaluate predictors of Covid-19 severity previously proposed.** Clinical and demographic characteristics and 120 single nucleotide polymorphisms were analyzed from 817 patients with Covid-19, who attended the emergency department of the Hospital Universitario de La Princesa (Madrid, Spain) during March and April 2020. The main outcome was a modified version of the 7-point World Health Organization (WHO) Covid-19 severity scale (WHOCS); both in the moment of the first hospital examination (WHOCS-1) and of the severest WHOCS score (WHOCS-2). ***TMPRSS2* rs75603675 genotype (OR=0.586), dyslipidemia (OR=2.289), sex (OR=0.586) and the Charlson Comorbidity Index (OR=1.126) were identified as the main predictors**

of disease severity. Consequently, these variables might influence Covid-19 severity and could be used as predictors of disease development.”

2. Introduction: When stating "Although several studies were published to date evaluating genetic biomarkers associated with Covid-19 severity" cite relevant papers and mention some examples, if possible.

Answer: thank you very much for your suggestion. We now mention relevant papers that reported several associations:

Original version (line 69):

“Although several studies were published to date evaluating genetic biomarkers associated with Covid-19 severity, most were exploratory, showing heterogenic results and, still nowadays, a clinically relevant genetic biomarker was not described. Consequently, confirmatory studies are warranted to determine the clinical relevance of the discovered markers. The aim of this work was to perform a review of the published single nucleotide polymorphisms (SNPs) related to Covid-19 prognosis or severity by the end of 2020 and to evaluate them in an independent validation cohort. For this purpose, we genotyped 817 patients managed at Hospital Universitario de La Princesa, Madrid, Spain, for a panel of 120 SNPs selected based on an extensive literature search.”

Modified version (line 73):

“Although several studies were published to date evaluating genetic biomarkers associated with Covid-19 severity, most were exploratory, showing heterogenic results and, still nowadays, a clinically relevant genetic biomarker was not described. **The first were genes involved in virus entrance to the host, such as angiotensin converting enzyme 2 gene (*ACE2*) and transmembrane serine protease 2 gene (*TMPRSS2*).** Different research groups have suggested several candidate variants of *ACE2*, namely rs2285666 or rs4646116 [6,7]. On the other hand, for the *TMPRSS2*, variants such as rs2298659, rs17854725, rs12329760 and rs75603675 were found to be different in the frequency in populations more affected by the disease [8,9]. However, a later Genome-Wide Association Study (GWAS) of severe Covid-19 with respiratory failure reported two clusters of genes associated with two different polymorphisms: rs11385942 in leucine zipper transcription factor like 1 gene (*LZTFL1*) and rs657152 in *ABO* gene [10]. Due to the disparity of the observed findings, additional confirmatory and exploratory studies are warranted. The aim of this work was to perform a review of the published single nucleotide polymorphisms (SNPs)...”

3. Methods: "Checkpoints were applied during recruitment to prioritize the selection of under-represented patients" should be explained in detail. Which patients do you consider underrepresented and why is it relevant to the study? Do you mean race, ethnic background, or socioeconomic status? Did you add this information as covariate to the statistical analysis? What are the checkpoints during recruitment?

Answer: thank you very much for your suggestion. As the recruitment was retrospective and consecutive, it could lead to a non-uniform representation of severities. Consequently, we checked the groups in order to assure a realistic number of patients within each grade of severity. Race and ethnic background or socio-economic status were not considered on the checkpoints because there were no very much variability in our population. The only selection variable was severity. The race and ethnic background was later considered as a variable in the analysis. We have included this explanation in Material and Methods as follows:

Original version (line 82):

“To balance the number of patients in each Covid-19 severity group, checkpoints were applied during recruitment to prioritize the selection of under-represented patients. The Ethics Research Committee of Hospital Universitario de La Princesa approved the study protocol. All subjects provided informed consent, except for the deceased.”

Modified version (line 94):

“To avoid imbalances in the proportions of the severity groups due to the retrospective and consecutive nature of recruitment, checkpoints were applied to prioritize the selection of under-represented patients, i.e. severe and mild patients. The first checkpoint was performed at the beginning of May 2021, with 617 patients recruited: 83 (13.5%) mild, 466 (75.5%) moderate and 68 (11%) severe; and the second was performed at the end of May 2021, with 743 patients: 159 (21.5%) mild, 502 (67.5%) moderate and 82 (11%) severe. The Ethics Research Committee of Hospital Universitario de La Princesa approved the study protocol...”

Original version (line 103):

“As independent variables or covariates, the following ones were analyzed: demographic characteristics (age, sex), comorbidity (obesity, dyslipidemia, tobacco or alcohol use, and the Charlson comorbidity index, CCI, relevant drugs used prior to Covid-19 infection i.e., angiotensin converter enzyme inhibitors (ACEIs) (e.g., enalapril, lisinopril), angiotensin receptor II antagonists (ARA-II) (e.g., losartan, irbesartan), aldosterone antagonists (e.g., eplerenone, spironolactone), oral anticoagulants (e.g., acenocumarol, dabigatran), systemic corticosteroids (e.g., dexamethasone) or systemic immunosuppressants (e.g., tacrolimus or methotrexate), drugs for Covid-19 treatment

(hydroxychloroquine or chloroquine, remdesivir, corticosteroids, tocilizumab, heparin, lopinavir/ritonavir, plasma transfusion) and SNPs.”

Modified version (line 115):

“As independent variables or covariates, the following ones were analyzed: demographic characteristics (age, sex), comorbidity (obesity, dyslipidemia, tobacco or alcohol use, and the Charlson comorbidity index, CCI), **biogeographical group [12] (inferred from the country of birth)**, relevant drugs used prior to Covid-19 infection i.e., angiotensin converter enzyme inhibitors (ACEIs) (e.g., enalapril, lisinopril), angiotensin receptor II antagonists (ARA-II) (e.g., losartan, irbesartan), aldosterone antagonists (e.g., eplerenone, spironolactone), oral anticoagulants (e.g., acenocumarol, dabigatran), systemic corticosteroids (e.g., dexamethasone) or systemic immunosuppressants (e.g., tacrolimus or methotrexate), drugs for Covid-19 treatment (hydroxychloroquine or chloroquine, remdesivir, corticosteroids, tocilizumab, heparin, lopinavir/ritonavir, plasma transfusion) and SNPs.”

Original version (line 143):

“Statistical analysis was conducted in R [87]. To analyze Severity 1 and 2 variables, a univariate analysis was initially performed by ordinal logistic regression with the MASS package [88]. As independent variables, the following ones were explored: genetic variables (all SNPs were transformed into *dummy* variables, that is, grouping heterozygous subjects with the most frequent homozygous diplotype), previous treatments and other comorbidities (e.g., dyslipidemia or tobacco use); CCI and sex were included as covariates and the level of significance was adjusted based on the Bonferroni correction for multiple comparisons. Those variables with corrected $p' < 0.05$ were introduced as independent variables in a multiple logistic regression analysis of Severity 1 and 2, i.e., the multivariate analysis. In this analysis, significance was similarly adjusted based on the Bonferroni correction for multiple comparisons.”

Modified version (line 158):

“Statistical analysis was conducted in R [87]. To analyze Severity 1 and 2 variables, a univariate analysis was initially performed by ordinal logistic regression with the MASS package [88]. As independent variables, the following ones were explored: genetic variables (all SNPs were transformed into *dummy* variables, that is, grouping heterozygous subjects with the most frequent homozygous diplotype), **biogeographic group**, previous treatments and other comorbidities (e.g., dyslipidemia or tobacco use);

CCI and sex were included as covariates and the level of significance was adjusted based on the Bonferroni correction for multiple comparisons. Those variables with corrected $p' < 0.05$ were introduced as independent variables in a multiple logistic regression analysis of Severity 1 and 2, i.e., the multivariate analysis. In this analysis, significance was similarly adjusted based on the Bonferroni correction for multiple comparisons.”

4. Please add more details on sample collection. Did you retrieve samples from the patients at the Intensive Care Unit when they arrived?

Answer: sample collection occurred in a period after discharge of the patients, during the months of January to May 2021. Patients who gave informed consent provided 5 ml of blood collected in an EDTA K2 tube and samples of deceased patients were retrieved from stored collections. We have added a paragraph providing further insight on sample extraction:

Original version (line 116):

“Finally, variants related to safety and effectiveness of the drugs used for the treatment of the disease were included, i.e. pharmacogenetic variants. Genomic DNA was extracted from peripheral blood samples with a Maxwell RSC automated DNA extractor (Promega) following the manufacturer’s instructions.”

Modified version (line 128):

“Finally, variants related to safety and effectiveness of the drugs used for the treatment of the disease were included, i.e. pharmacogenetic variants. **Sample collection occurred in a period after discharge of the patients, during the months of January to May 2021. Patients who gave informed consent provided 5 ml of blood collected in an EDTA K2 tube. For deceased patients, samples were retrieved from stored collection.** Genomic DNA was extracted from peripheral blood samples with a Maxwell RSC automated DNA extractor (Promega) following the manufacturer’s instructions.”

5. Results: Please add more details on the individuals. In the methods section, you mention allele frequencies for the SNPs for Hispanics and Iberians. However, you do not explain the ethnic/racial background of the individuals in the results section. Moreover, based on the different allele frequencies in individuals with different ancestry, even if the genetic ancestry is not used, as it is not measured in this study, the racial background could be included as covariate. The significant results might be due to the differing allele frequencies between the different ancestry groups (Iberians are of European, and Americans are of Hispanic (admixed) ancestry). Please state both as limitations in the limitations section.

Answer: thank you very much for your suggestion. As stated in the second comment, race was indeed considered. However, in our opinion, race or ancestry or ethnic background are imprecise terms, which derive from other variables (e.g., genetics or biogeographic origin). Therefore, to our understanding, the closest way to address “race”, is with the biogeographic group [PMID: 30506572], which we estimated with the country of origin. In this study, the univariate analysis did not reveal any significant relationship concerning the biogeographic origin. For this reason, this information was not shown in the original version of the manuscript. Nonetheless, thanks to this comment, we realize now that further insight is needed regarding the biogeographical group of origin, so we have included this in the main text:

Original version (line 103):

“As independent variables or covariates, the following ones were analyzed: demographic characteristics (age, sex), comorbidity (obesity, dyslipidemia, tobacco or alcohol use, and the Charlson comorbidity index, CCI, relevant drugs used prior to Covid-19 infection i.e., angiotensin converter enzyme inhibitors (ACEIs) (e.g., enalapril, lisinopril), angiotensin receptor II antagonists (ARA-II) (e.g., losartan, irbesartan), aldosterone antagonists (e.g., eplerenone, spironolactone), oral anticoagulants (e.g., acenocumarol, dabigatran), systemic corticosteroids (e.g., dexamethasone) or systemic immunosuppressants (e.g., tacrolimus or methotrexate), drugs for Covid-19 treatment (hydroxychloroquine or chloroquine, remdesivir, corticosteroids, tocilizumab, heparin, lopinavir/ritonavir, plasma transfusion) and SNPs.”

Modified version (line 115):

“As independent variables or covariates, the following ones were analyzed: demographic characteristics (age, sex), comorbidity (obesity, dyslipidemia, tobacco or alcohol use, and the Charlson comorbidity index, CCI), **biogeographical group [12] (inferred from the country of birth)**, relevant drugs used prior to Covid-19 infection i.e., angiotensin converter enzyme inhibitors (ACEIs) (e.g., enalapril, lisinopril), angiotensin receptor II antagonists (ARA-II) (e.g., losartan, irbesartan), aldosterone antagonists (e.g., eplerenone, spironolactone), oral anticoagulants (e.g., acenocumarol, dabigatran), systemic corticosteroids (e.g., dexamethasone) or systemic immunosuppressants (e.g., tacrolimus or methotrexate), drugs for Covid-19 treatment (hydroxychloroquine or chloroquine, remdesivir, corticosteroids, tocilizumab, heparin, lopinavir/ritonavir, plasma transfusion) and SNPs.”

Original version (line 143):

“Statistical analysis was conducted in R [87]. To analyze Severity 1 and 2 variables, a univariate analysis was initially performed by ordinal logistic regression with the MASS package [88]. As independent variables, the following ones were explored: genetic variables (all SNPs were transformed into *dummy* variables, that is, grouping heterozygous subjects with the most frequent homozygous diplotype), previous treatments and other comorbidities (e.g., dyslipidemia or tobacco use); CCI and sex were included as covariates and the level of significance was adjusted based on the Bonferroni correction for multiple comparisons. Those variables with corrected $p' < 0.05$ were introduced as independent variables in a multiple logistic regression analysis of Severity 1 and 2, i.e., the multivariate analysis. In this analysis, significance was similarly adjusted based on the Bonferroni correction for multiple comparisons.”

Modified version (line 158):

“Statistical analysis was conducted in R [87]. To analyze Severity 1 and 2 variables, a univariate analysis was initially performed by ordinal logistic regression with the MASS package [88]. As independent variables, the following ones were explored: genetic variables (all SNPs were transformed into *dummy* variables, that is, grouping heterozygous subjects with the most frequent homozygous diplotype), **biogeographic group**, previous treatments and other comorbidities (e.g., dyslipidemia or tobacco use); CCI and sex were included as covariates and the level of significance was adjusted based on the Bonferroni correction for multiple comparisons. Those variables with corrected $p' < 0.05$ were introduced as independent variables in a multiple logistic regression analysis of Severity 1 and 2, i.e., the multivariate analysis. In this analysis, significance was similarly adjusted based on the Bonferroni correction for multiple comparisons.”

Original version (line 159):

“The population consisted on 817 patients, 453 (55.45%) males and 364 (44.55%) females. The range of age was 19 to 97 years-old, where the mean age was 60 years-old. The baseline characteristics of the study population are shown in Table 2.”

Modified version (line 175):

“The population consisted on 817 patients, 453 (55.45%) males and 364 (44.55%) females. The range of age was 19 to 97 years-old, where the mean age was 60 years-old. Biogeographical origin of patients was inferred by their country of birth: 636 were European, 161 were American, 7 were East Asian, 6 were Near Eastern and 1 was Central/South Asian. The baseline characteristics of the study population are shown in Table 2.”

Original version (line 183):

“The univariate analysis of Severity 1 and 2 variables is shown in Supplementary Table 1A, including a summary of nominally significant variables and those who reached a corrected $p < 0.05$, which were included in the multivariate analysis. Males compared to females, a higher CCI (covariates), dyslipidemia and TMPRSS2 rs75603675 (C/C or C/A diplotypes, compared to the A/A diplotype) were significantly related to a higher Severity 1 and 2 status, after multivariate analysis and Bonferroni correction for multiple comparisons (Figure 1).”

Modified version (line 197):

“The univariate analysis of Severity 1 and 2 variables is shown in Supplementary Table 1A, including a summary of nominally significant variables and those who reached a corrected $p < 0.05$, which were included in the multivariate analysis. Biogeographical group resulted non-significant. Males compared to females (OR = 0.586), a higher CCI (OR = 1.126) (covariates), dyslipidemia (OR = 2.289) and TMPRSS2 rs75603675 (C/C or C/A diplotypes, compared to the A/A diplotype) (OR =2.140) were significantly related to a higher Severity 1 and 2 status, after multivariate analysis and Bonferroni correction for multiple comparisons (Figure 1).”

Additionally, we have included these results in Supplementary Table 1 along with the other non-significant results:

Supplementary Table 1. Summary of nominally significant associations and whole results of the univariate analysis of A) the Severity 1 and Severity 2 variables and B) the WHOCS-1 and WHOCS-2 variables.

A)

	Estimate	SE	p		Estimate	SE	p
(Intercept)	3.212	0.057	0.000	(Intercept)	3.436	0.078	0.000
Immunosuppressants	-0.080	0.193	0.680	Immunosuppressants	-0.129	0.265	0.627

Sex	-0.201	0.062	0.001	Sex	-0.371	0.085	0.000
CCI	0.072	0.014	0.000	CCI	0.123	0.020	0.000
	Estimate	SE	p		Estimate	SE	p
(Intercept)	2.884	0.086	0.000	(Intercept)	3.120	0.106	0.000
Biogeographic group	0.157	0.108	0.335	Biogeographic group	0.128	0.133	0.335
Sex	-0.308	0.089	0.001	Sex	-0.483	0.110	0.000
CCI	0.104	0.021	0.000	CCI	0.155	0.025	0.000

6. Please add odds ratios to the text to see the direction of the effects when explaining the results of Figure 1.

Answer: thank you very much for your suggestion. We have added the odds ratios in the text and in the abstract as follows:

Original version (line 41):

“By the end of December 2021, coronavirus disease 2019 (Covid-19) produced more than 271 million cases and 5.3 million deaths. Although vaccination is proved an effective strategy for pandemic control, it is not yet equally available in all countries over the world. Therefore, the authorization of effective therapies and identification of prognostic biomarkers remains crucial to manage Covid-19 patients more rationally. Clinical and demographic characteristics and 120 single nucleotide polymorphisms were analyzed from 817 patients with Covid-19, who attended the emergency department of the Hospital Universitario de La Princesa (Madrid, Spain) during March and April 2020. The main outcome was a modified version of the 7-point world health organization (WHO) Covid-19 severity scale (WHOCS); both in the moment of the first hospital examination (WHOCS-1) and of the severest WHOCS score (WHOCS-2). *TMPRSS2* rs75603675 genotype, dyslipidemia, sex and the Charlson Comorbidity Index were identified as the main predictors of disease severity. Afterwards, the integration of all these variables into simple equations was performed.”

Modified version (line 41):

“By the end of December 2021, coronavirus disease 2019 (Covid-19) produced more than 271 million cases and 5.3 million deaths. **Although vaccination was an effective strategy for pandemic control, it is not yet equally available in all countries. Therefore, identification of prognostic biomarkers remains crucial to manage Covid-19 patients. The aim of this study was to evaluate predictors of Covid-19 severity previously proposed.** Clinical and demographic characteristics and 120 single nucleotide polymorphisms were analyzed from 817 patients with Covid-19, who attended the emergency department of the Hospital Universitario de La Princesa (Madrid, Spain) during March and April 2020. The main

outcome was a modified version of the 7-point World Health Organization (WHO) Covid-19 severity scale (WHOCS); both in the moment of the first hospital examination (WHOCS-1) and of the severest WHOCS score (WHOCS-2). **TMPRSS2 rs75603675 genotype (OR=0.586), dyslipidemia (OR=2.289), sex (OR=0.586) and the Charlson Comorbidity Index (OR=1.126) were identified as the main predictors of disease severity. Consequently, these variables might influence Covid-19 severity and could be used as predictors of disease development"**

Original version (line 183):

"The univariate analysis of Severity 1 and 2 variables is shown in Supplementary Table 1A, including a summary of nominally significant variables and those who reached a corrected $p < 0.05$, which were included in the multivariate analysis. Males compared to females, a higher CCI (covariates), dyslipidemia and TMPRSS2 rs75603675 (C/C or C/A diplotypes, compared to the A/A diplotype) were significantly related to a higher Severity 1 and 2 status, after multivariate analysis and Bonferroni correction for multiple comparisons (Figure 1)."

Modified version (line 197):

"The univariate analysis of Severity 1 and 2 variables is shown in Supplementary Table 1A, including a summary of nominally significant variables and those who reached a corrected $p < 0.05$, which were included in the multivariate analysis. **Biogeographical group resulted non-significant.** Males compared to females (**OR = 0.586**), a higher CCI (**OR = 1.126**) (covariates), dyslipidemia (**OR = 2.289**) and TMPRSS2 rs75603675 (C/C or C/A diplotypes, compared to the A/A diplotype) (**OR = 2.140**) were significantly related to a higher Severity 1 and 2 status, after multivariate analysis and Bonferroni correction for multiple comparisons (Figure 1)."

7. Discussion: Do you think that the different variants (delta, omicron, etc.) since you finished sample collection (April 2020) would have different predictors? Please discuss the possibility after "Firstly, patients were recruited from the first Covid-19 wave". Please explain what the original finding with TMPRSS2 was, in the manuscript where you collected the information on the gene from.

Answer: thank you very much for your interesting comment and suggestion. We acknowledge that the different Sars-CoV-2 variants may lead to different predictors and different genetic susceptibilities. A vast amount of works studying the genetic susceptibility to Covid-19 used data from the first virus strand, which justified conducting the present work. However, we agree that additional studies are warranted to validate our results in patients infected with other strains. As a matter of fact, we are designing a validation study of the results of this

research work with 2000 additional patients of the first wave and 4000 patients of the rest waves, which were infected with other virus strands. Our theory is that this biomarker may have more predictive power in variants such as omicron, which is more infectious, and this variant apparently facilitates entry of the virus into the host cell. We rearranged the text as you request to enhance the importance of these aspects:

Original version (line 281):

“Firstly, patients were recruited from the first Covid-19 wave. That means that hospital protocols were severely affected by the emergency healthcare situation, and adequacy of the therapeutic effort was needed. This, in combination with the retrospective nature of the study produces scarcity of deceased and asymptomatic individuals. Consequently, the severity distribution might be skewed. Another limitation is the nature of the WHOCS-1 and WHOCS-2 analysis by means of the generalized liner model, which assumes a normal distribution of these variables; in contrast, the fact that the exact same variables related to Severity 1 and 2 variables were identified is reassuring, along with the strict control for type-1 error.”

Modified version (line 304):

“Firstly, patients were recruited from the first Covid-19 wave, which might be considered more than a limitation; it can be considered a strength. As this work was carried with patients infected during March-April 2020, the circulating strains were different from those circulating now. In this sense, the conclusions regarding *TMPRSS2* rs75603675 should be confirmed in the currently circulating strains. Nonetheless, new strains, such as the omicron variant, could be more infectious or pathogenic, being the underlying mechanism of such pathogenicity an enhanced interaction between viral antigens and *TMPRSS2*. This interaction could be affected by genetic variants of this gene and cause an even greater difference of severity with new strains. Nevertheless, new studies shall confirm the relevance of *TMPRSS2* rs75603675 in new circulating variants. Another limitation is that hospital protocols were severely affected by the emergency healthcare situation of the first Covid-19 wave, and adequacy of the therapeutic effort was needed. This, in combination with the retrospective nature of the study produces relative scarcity of severe and asymptomatic individuals. Consequently, the severity distribution might be skewed. In addition, the nature of the WHOCS-1 and WHOCS-2 analysis by means of the generalized linear model assumes a normal distribution of these variables. To avoid bias, we performed a strict statistical analysis, in exchange for assuming greater Type II error. Moreover, the fact that the exact same variables related to Severity 1 and 2 variables were identified is reassuring, along with the strict control for type-1 error.

In addition, we now discuss the original source from which we extracted the polymorphism *TPMRSS2* rs75603675:

Original version (line 250):

“Probably, the reason behind this association is explained by the downregulation of the protein in *TPMRSS2* rs75603675 (G>A) A allele carriers, or the expression of a protein with a structural change that causes a less specific or impaired interaction with viral proteins, causing a less efficient internalization of viruses inside human cells.

Other nominally significant associations were established between WHOCS scales and: IFNL4 rs12979860, ACEIs, obesity, ARA-II, HCP5 rs2395029, ACE rs1799752, DPP4 rs17574, HMOX1 rs2071746, IL10 rs1800896, IL6 rs1818879, NFKB1 rs28362491 and MX1 rs469390.”

Modified version (line 268):

“Probably, the reason behind this association is explained by the downregulation of the protein in *TPMRSS2* rs75603675 (G>A) A allele carriers, or the expression of a protein with a structural change that causes a less specific or impaired interaction with viral proteins, causing a less efficient internalization of viruses inside human cells. **This is congruent with Latini et al observations: *TPMRSS2* rs75603675 (G>A) A allele was in significantly lower frequency in populations that suffered more severe cases of Covid-19 [76]. This suggested a possible protective effect towards Covid-19 infection.”**

Other nominally significant associations were established between WHOCS scales and: IFNL4 rs12979860, ACEIs, obesity, ARA-II, HCP5 rs2395029, ACE rs1799752, DPP4 rs17574, HMOX1 rs2071746, IL10 rs1800896, IL6 rs1818879, NFKB1 rs28362491 and MX1 rs469390.”

8. In general, the manuscript is easy to read, but please try to form more technical sentences on some occasions, e.g., "With this work, we link several pieces of the puzzle in place".

Answer: thank you very much for your suggestion. We have thoughtfully revised the text and applied a more technical language.

Reviewer #2

1. Page 10, line 181: "Males compared to females, a higher CCI (covariates), dyslipidemia and TMPRSS2 rs75603675 (C/C or C/A diplotypes, compared to the A/A diplotype) were significantly related to a higher Severity 1 and 2 status". This finding is not reflected in Figure 1, where it appears as if males have a lower severity status.

Answer: apologies for this mistake, we have corrected the typo in the Figure 1:

Original version:

Modified version:

2. Table 2 and 4: The abbreviations WHOCS-1 and 2 are used, whereas in Figure 1 Severity 1 and 2 is used. This is confusing.

Answer: "Severity" is a categorical variable whereas "WHOCS" is a continuous variable. Using a categorical variable in first place allowed a more restrictive analysis to robustly confirm the influence of predictors on disease severity, based on relative risks. Afterwards, we used the

related variable WHOCS to provide a quantitative measurement of severity, to propose the equations that may be more clinically implementable.

3. Table 5: the brackets for (0.101xCCI) can be removed as this is confusing

Answer: we have removed the brackets in order to be clearer:

Original version (line 221):

Table 5. Equations for the prediction of WHOCS-1 and WHOCS-2 for Covid-19 severity of patients.

	1.962	Basal severity
	+ 0.579	If patient has dyslipidemia
WHOCS-1	+ 0.591	If TMPRSS2 rs75603675 C/C or C/A genotypes are present
	+ (0.101xCCI)	
	- 0.326	If patient is female
Total:		

	2.597	Basal severity
	+ 0.551	If patient has dyslipidemia
WHOCS-2	+ 0.405	If TMPRSS2 rs75603675 C/C or C/A genotypes are present
	+ (0.120xCCI)	
	- 0.415	If patient is female
Total:		

Modified version (line 221):

Table 5. Equations for the prediction of WHOCS-1 and WHOCS-2 for Covid-19 severity of patients.

	1.962	Basal severity
	+ 0.579	If patient has dyslipidemia
WHOCS-1	+ 0.591	If TMPRSS2 rs75603675 C/C or C/A genotypes are present
	+ 0.101	x CCI
	- 0.326	If patient is female
Total:		

	2.597	Basal severity
	+ 0.551	If patient has dyslipidemia
WHOCS-2	+ 0.405	If TMPRSS2 rs75603675 C/C or C/A genotypes are present
	+ 0.120	x CCI
	- 0.415	If patient is female
Total:		

4. Page 15, line 279: generalized linear model

Answer: thank you very much for your suggestion. We have emended the misspelling:

Original version (line 274):

“Another limitation is the nature of the WHOCS-1 and WHOCS-2 analysis by means of the generalized liner model, which assumes a normal distribution of these variables; in contrast, the fact that the exact same variables related to Severity 1 and 2 variables were identified is reassuring, along with the strict control for type-1 error.”

Modified version (line 316):

“**In addition**, the nature of the WHOCS-1 and WHOCS-2 analysis by means of the generalized **linear** model assumes a normal distribution of these variables. **To avoid bias, we performed a strict statistical analysis, in exchange for assuming greater Type II error. Moreover**, the fact that the exact same variables related to Severity 1 and 2 variables were identified is reassuring, along with the strict control for type-1 error.”

Reviewer #3

1. The absence of a control group from the same community - I note that the authors quote Iberian/US frequencies for the SNPs of interest - were these detected using the same platform. Is the lack of direct comparison by this group a concern?

Answer: the allele frequencies are cited from bibliography, as we expected a majority of Iberian and Latin population. Consequently, we tested biogeographic group as a variable. As the results were not significant, we omitted this information. Nonetheless, as there are some differences in allele distributions between biogeographic groups, we repeated the analysis including it as a covariable. The results remained consistent and the biogeographic group was still non-significant. We have included it now in the results section as follows:

Original version (line 103):

“As independent variables or covariates, the following ones were analyzed: demographic characteristics (age, sex), comorbidity (obesity, dyslipidemia, tobacco or alcohol use, and the Charlson comorbidity index, CCI, relevant drugs used prior to Covid-19 infection i.e., angiotensin converter enzyme inhibitors (ACEIs) (e.g., enalapril, lisinopril), angiotensin receptor II antagonists (ARA-II) (e.g., losartan, irbesartan), aldosterone antagonists (e.g., eplerenone, spironolactone), oral anticoagulants (e.g., acenocumarol, dabigatran), systemic corticosteroids (e.g., dexamethasone) or systemic immunosuppressants (e.g., tacrolimus or methotrexate), drugs for Covid-19 treatment (hydroxychloroquine or chloroquine, remdesivir, corticosteroids, tocilizumab, heparin, lopinavir/ritonavir, plasma transfusion) and SNPs.”

Modified version (line 118):

“As independent variables or covariates, the following ones were analyzed: demographic characteristics (age, sex), comorbidity (obesity, dyslipidemia, tobacco or alcohol use, and the Charlson comorbidity index, CCI), **biogeographical group [12] (inferred from the country of birth)**, relevant drugs used prior to Covid-19 infection i.e., angiotensin converter enzyme inhibitors (ACEIs) (e.g., enalapril, lisinopril), angiotensin receptor II antagonists (ARA-II) (e.g., losartan, irbesartan), aldosterone antagonists (e.g., eplerenone, spironolactone), oral anticoagulants (e.g., acenocumarol, dabigatran), systemic corticosteroids (e.g., dexamethasone) or systemic immunosuppressants (e.g., tacrolimus or methotrexate), drugs for Covid-19 treatment (hydroxychloroquine or chloroquine, remdesivir, corticosteroids, tocilizumab, heparin, lopinavir/ritonavir, plasma transfusion) and SNPs.”

Original version (line 143):

“Statistical analysis was conducted in R [87]. To analyze Severity 1 and 2 variables, a univariate analysis was initially performed by ordinal logistic regression with the MASS package [88]. As independent variables, the following ones were explored: genetic variables (all SNPs were transformed into *dummy* variables, that is, grouping heterozygous subjects with the most frequent homozygous diplotype), previous treatments and other comorbidities (e.g., dyslipidemia or tobacco use); CCI and sex were included as covariates and the level of significance was adjusted based on the Bonferroni correction for multiple comparisons. Those variables with corrected $p' < 0.05$ were introduced as independent variables in a multiple logistic regression analysis of Severity 1 and 2, i.e., the multivariate analysis. In this analysis, significance was similarly adjusted based on the Bonferroni correction for multiple comparisons.”

Modified version (line 161):

“Statistical analysis was conducted in R [87]. To analyze Severity 1 and 2 variables, a univariate analysis was initially performed by ordinal logistic regression with the MASS package [88]. As independent variables, the following ones were explored: genetic variables (all SNPs were transformed into *dummy* variables, that is, grouping heterozygous subjects with the most frequent homozygous diplotype), **biogeographic group**, previous treatments and other comorbidities (e.g., dyslipidemia or tobacco use); CCI and sex were included as covariates and the level of significance was adjusted based on the Bonferroni correction for multiple comparisons. Those variables with corrected $p' < 0.05$ were introduced as independent variables in a multiple logistic regression analysis of Severity 1 and 2, i.e., the multivariate analysis. In this analysis, significance was similarly adjusted based on the Bonferroni correction for multiple comparisons.”

Original version (line 159):

“The population consisted on 817 patients, 453 (55.45%) males and 364 (44.55%) females. The range of age was 19 to 97 years-old, where the mean age was 60 years-old. The baseline characteristics of the study population are shown in Table 2.”

Modified version (line 178):

“The population consisted on 817 patients, 453 (55.45%) males and 364 (44.55%) females. The range of age was 19 to 97 years-old, where the mean age was 60 years-old. Biogeographical origin of patients was inferred by their country of birth: 636 were European, 161 were American, 7 were East Asian, 6 were Near Eastern and 1 was Central/South Asian. The baseline characteristics of the study population are shown in Table 2.”

Original version (line 183):

“The univariate analysis of Severity 1 and 2 variables is shown in Supplementary Table 1A, including a summary of nominally significant variables and those who reached a corrected $p < 0.05$, which were included in the multivariate analysis. Males compared to females, a higher CCI (covariates), dyslipidemia and TMPRSS2 rs75603675 (C/C or C/A diplotypes, compared to the A/A diplotype) were significantly related to a higher Severity 1 and 2 status, after multivariate analysis and Bonferroni correction for multiple comparisons (Figure 1).”

Modified version (line 200):

“The univariate analysis of Severity 1 and 2 variables is shown in Supplementary Table 1A, including a summary of nominally significant variables and those who reached a corrected $p < 0.05$, which were included in the multivariate analysis. Biogeographical group resulted non-significant. Males compared to females (OR = 0.586), a higher CCI (OR = 1.126) (covariates), dyslipidemia (OR = 2.289) and TMPRSS2 rs75603675 (C/C or C/A diplotypes, compared to the A/A diplotype) (OR =2.140) were significantly related to a higher Severity 1 and 2 status, after multivariate analysis and Bonferroni correction for multiple comparisons (Figure 1).”

Additionally, we have included these results in Supplementary Table 1 along with the other non-significant results:

Supplementary Table 1. Summary of nominally significant associations and whole results of the univariate analysis of A) the Severity 1 and Severity 2 variables and B) the WHOCS-1 and WHOCS-2 variables.

A)

	Estimate	SE	p		Estimate	SE	p
(Intercept)	3.212	0.057	0.000	(Intercept)	3.436	0.078	0.000
Immunosuppressants	-0.080	0.193	0.680	Immunosuppressants	-0.129	0.265	0.627

Sex	-0.201	0.062	0.001	Sex	-0.371	0.085	0.000
CCI	0.072	0.014	0.000	CCI	0.123	0.020	0.000
	Estimate	SE	p		Estimate	SE	p
(Intercept)	2.884	0.086	0.000	(Intercept)	3.120	0.106	0.000
Biogeographic group	0.157	0.108	0.335	Biogeographic group	0.128	0.133	0.335
Sex	-0.308	0.089	0.001	Sex	-0.483	0.110	0.000
CCI	0.104	0.021	0.000	CCI	0.155	0.025	0.000

2. The relatively small number of individuals with severe disease - 106 if use groups 5-7. Given the large number of experimental treatments used in this cohort there appears to be significant potential to introduce bias. I would suggest comments from someone with real biostatistical expertise but the combination of no experimental controls, targeted SNP analysis and a relatively small 'severe' cohort does raise concerns regarding false positives.

Answer: we agree that the relatively small severe cohort is a limitation. That is the principal reason why we applied such a strict statistical analysis, prioritizing the control of type 1 error rather than type 2 error. In fact, we were interested in confirming very strong associations rather than discovering new ones with lower clinical relevance. Moreover, note this work is observational, not a clinical trial intended to demonstrate an effect with a specific statistical power. Alternatively, we designed it as a validation study of previously proposed biomarkers. Nonetheless, we have elaborated on these limitations as follows:

Original version (line 274):

“Despite the merits of this work, it presents limitations that should be considered. Firstly, patients were recruited from the first Covid-19 wave. That means that hospital protocols were severely affected by the emergency healthcare situation, and adequacy of the therapeutic effort was needed. This, in combination with the retrospective nature of the study produces scarcity of deceased and asymptomatic individuals. Consequently, the severity distribution might be skewed”

Modified version (line 313):

“Another limitation is that hospital protocols were severely affected by the emergency healthcare situation of the first Covid-19 wave, and adequacy of the therapeutic effort was needed. This, in combination with the retrospective nature of the study produces relative scarcity of severe and asymptomatic individuals. Consequently, the severity distribution might be skewed. In addition, the nature of the WHOCS-1 and WHOCS-2 analysis by means of the generalized linear model assumes a normal distribution of these variables. To avoid bias, we performed a strict statistical analysis, in exchange for assuming greater Type II error. Moreover, the fact that the exact same variables related to Severity 1 and 2 variables were identified is reassuring, along with the strict control for type-1 error.”

3. If I understand correctly age and obesity did not predict outcome but sex did. This is contrary to much larger clinical series e.g. the OpenSAFELY study - Williamson et al 2020. Can the authors explain why? Does this again raise concerns regarding the cohort?

Answer: In our study, age and obesity do predict outcome maybe because those parameters are included in Charlson Comorbidity Index. We performed the analysis with CCI to simplify all the factors that were already proven to have an effect (i.e. age, obesity, respiratory comorbidities, etc.). Nonetheless, we understand that this is not clear in the text. Therefore, we have emphasized that in the text:

Original version (line 216):

“Secondly, we proposed to correct for confounding factors in all the statistical tests performed, and therefore decided to consider the CCI and the sex as covariates.”

Modified version (line 236):

“Secondly, we proposed to correct for confounding factors in all the statistical tests performed, and therefore decided to consider the CCI, **which included known COVID-19 severity predictors such as age and obesity**, and the sex as covariates”

4. There have been many many studies of SNPs in COVID including some of severity, including one from Madrid: DOI: 10.1056/NEJMoa2020283 They did not implicate this SNP. Was it tested for. How do the authors explain the prior lack of signal at this locus?

Answer: the principal reason is that our study design and genotyping array design was performed before the release of that study, and that we observed previous evidence by other authors (i.e., Latini et al.). The implemented whole-genome coverage (10.1056/NEJMoa2020283) should have included the *TMPRSS2* SNP region. The mentioned study is one of the most reliable and robust studies on the field to date. The explanations that we can come across to address the discrepancy is that our outcome variable is different. Our analysis focus in all Covid-19 severity stages, whereas they focus in the most severe cases. Thus, our findings may explain the susceptibility to infection part of the disease whereas other factors may determine terminal status of Covid-19. Definitely, more research must be performed in other to clarify the true effect of the polymorphism in the outcome of this disease. We understand the relevance of the discussion of this topic, given the magnitude of the GWAS study. Consequently, we have included this reflection in the text:

Original version (line 259):

“...Hence, we considered all associations mentioned above as negative, and not worth discussing, because this would contribute to confusion.

With this work, we link several pieces of the puzzle in place. A variable that excellently captures the comorbidity of patients is used, the CCI index...”

Modified version (line 282):

“...Hence, we considered all associations mentioned above as negative, and not worth discussing, because this would contribute to confusion.

One noteworthy disparity with our study is the GWAS study by The Severe Covid-19 GWAS Group [14]. They report different genetic markers to ours. The explanation that we can elucidate to address this discrepancy is that our outcome variable is different. Our analysis focus in all Covid-19 severity stages, whereas they focus in the most severe cases. Thus, our findings may explain the susceptibility to infection in the early stages of the disease, whereas other factors such as 3p21.31 gene cluster and ABO blood-group system may determine terminal status of Covid-19. Definitely, more research must be performed in order to clarify the true effect of the *TMPRSS2* rs75603675 polymorphism in the outcome of this disease.

With this work, we integrate multiple clinical factors and genetic factors to COVID-19 severity prediction. A variable that excellently captures the comorbidity of patients is used, the CCI index...”

5. There are a number of minor issues including the labelling of the charts regarding male-female risk which to my reading suggest an excess risk for female sex as presented whereas the text is clear that the opposite is true.

Answer: apologies for this mistake, we have corrected the typo in the Figure 1:

Original version:

Modified version:

May 5, 2022

RE: Life Science Alliance Manuscript #LSA-2022-01396-TR

Mr. Pablo Zubiaur

Clinical Pharmacology Department, Hospital Universitario La Princesa, Instituto Teofilo Hernando, Universidad Autonoma de Madrid (UAM), Instituto de Investigacion Sanitaria La Princesa (IIS-IP), CIBERehd, Madrid, Spain
C/Diego de León, 62
Madrid, Madrid 28006
Spain

Dear Dr. Zubiaur,

Thank you for submitting your revised manuscript entitled "TMPRSS2 rs75603675, comorbidity and sex are the primary predictors of Covid-19 severity.". We would be happy to publish your paper in Life Science Alliance pending final revisions necessary to meet our formatting guidelines.

- please add ORCID ID for all corresponding authors; you should have received instructions on how to do so
- please add the Twitter handle of your host institute/organization as well as your own or/and one of the authors in our system
- please consult our manuscript preparation guidelines <https://www.life-science-alliance.org/manuscript-prep> and make sure your manuscript sections are in the correct order
- please use the [10 author names, et al.] format in your references (i.e. limit the author names to the first 10)

A. FINAL FILES:

B. MANUSCRIPT ORGANIZATION AND FORMATTING:

Sincerely,

Reviewer #1 (Comments to the Authors (Required)):

The quality of the paper significantly improved after the revision. I have no more major concerns.

May 13, 2022

RE: Life Science Alliance Manuscript #LSA-2022-01396-TRR

Mr. Pablo Zubiaur

Clinical Pharmacology Department, Hospital Universitario La Princesa, Instituto Teofilo Hernando, Universidad Autonoma de Madrid (UAM), Instituto de Investigacion Sanitaria La Princesa (IIS-IP), CIBERehd, Madrid, Spain
C/Diego de León, 62
Madrid, Madrid 28006
Spain

Dear Dr. Zubiaur,

Thank you for submitting your Research Article entitled "TMPRSS2 rs75603675, comorbidity and sex are the primary predictors of Covid-19 severity.". It is a pleasure to let you know that your manuscript is now accepted for publication in Life Science Alliance. Congratulations on this interesting work.

DISTRIBUTION OF MATERIALS:

Again, congratulations on a very nice paper. I hope you found the review process to be constructive and are pleased with how the manuscript was handled editorially. We look forward to future exciting submissions from your lab.

Sincerely,
